# Using Machine Learning to Explore Shared Genetic Pathways and Possible Endophenotypes in Autism Spectrum Disorder

**DOI:** 10.3390/genes14020313

**Published:** 2023-01-25

**Authors:** Daniele Di Giovanni, Roberto Enea, Valentina Di Micco, Arianna Benvenuto, Paolo Curatolo, Leonardo Emberti Gialloreti

**Affiliations:** 1Department of Industrial Engineering, University of Rome Tor Vergata, Via del Politecnico 1, 00133 Rome, Italy; 2IMME Research Centre, Via San Francesco d’Assisi 20, 81100 Caserta, Italy; 3Child Neurology and Psychiatry Unit, Systems Medicine Department, University of Rome Tor Vergata, Via Montpellier 1, 00133 Rome, Italy; 4Department of Human Studies - Communication, Education, and Psychology, LUMSA University, Via di Borgo Sant Angelo 13, 00193 Rome, Italy; 5Department of Biomedicine and Prevention, University of Rome Tor Vergata, Via Montpellier 1, 00133 Rome, Italy

**Keywords:** Autism spectrum disorder (ASD), cluster analysis, gene networks, genotype–phenotype embedding, machine learning, patient similarity analytics, neurite morphogenesis, connectivity, neurobehavioral phenotypes, synapses, neurotransmission

## Abstract

Autism spectrum disorder (ASD) is a heterogeneous condition, characterized by complex genetic architectures and intertwined genetic/environmental interactions. Novel analysis approaches to disentangle its pathophysiology by computing large amounts of data are needed. We present an advanced machine learning technique, based on a clustering analysis on genotypical/phenotypical embedding spaces, to identify biological processes that might act as pathophysiological substrates for ASD. This technique was applied to the VariCarta database, which contained 187,794 variant events retrieved from 15,189 individuals with ASD. Nine clusters of ASD-related genes were identified. The 3 largest clusters included 68.6% of all individuals, consisting of 1455 (38.0%), 841 (21.9%), and 336 (8.7%) persons, respectively. Enrichment analysis was applied to isolate clinically relevant ASD-associated biological processes. Two of the identified clusters were characterized by individuals with an increased presence of variants linked to biological processes and cellular components, such as axon growth and guidance, synaptic membrane components, or transmission. The study also suggested other clusters with possible genotype–phenotype associations. Innovative methodologies, including machine learning, can improve our understanding of the underlying biological processes and gene variant networks that undergo the etiology and pathogenic mechanisms of ASD. Future work to ascertain the reproducibility of the presented methodology is warranted.

## 1. Introduction

Autism spectrum disorder (ASD) is a neurodevelopmental disorder characterized by deficits in social communication and interactions, and restrictive and repetitive patterns of behavior or interests. Its estimated prevalence is 1 in 59 children [1]. ASD presents with a substantial variability of clinical symptoms and a heterogeneous genetic architecture. Only a handful of ASD-related diseases have monogenic causes. This is, for example, the case of tuberous sclerosis complex (TSC), in which the dysregulation of the neurotransmission of GABA, resulting from genetic mutations of the mTOR pathway, has been established to underlie the development of both epilepsy and ASD in these individuals [2]. 

The disruption of different neurodevelopmental pathways associated with a relatively high number of genes makes it difficult to disentangle the exact mechanisms involved in ASD. Therefore, its genetic foundations still need to be further elucidated [3]. Nevertheless, progress in sequencing technology has improved the capability of identifying possible ASD risk genes, such as synaptic activity-related genes [4,5,6] as well as genes related to molecular regulatory systems [7,8,9], transcription and chromatin modeling [10,11], or the mTOR pathway [12]. Therefore, there is an urgent need to identify ASD-associated biomarkers and features—such as endophenotypes—to support diagnostics and to develop predictive ASD models [13].

Many approaches have been postulated to better understand these mechanisms. Machine learning algorithms have been widely applied in diagnostic tools for ASD. For example, Han adopted a novel evolutionary algorithm, the conjunctive clause evolutionary algorithm (CCEA), to select major features to better characterize individuals with ASD, thus demonstrating how machine learning tools might implement diagnostic models in ASD [13]. Kwon and colleagues predicted ASD symptom severity utilizing the fully automatic nodal feature extractor and the sparse hierarchical graph representation framework to encode the brain’s functional connectivity [14]. Ruther et al. trained random forest models on the Autism Diagnostic Observation Schedule (ADOS), a standardized diagnostic test for diagnosing and assessing ASD, to predict a diagnosis of ASD, while differentiating it from other neurodevelopmental disorders [15]. All these approaches underline the increasing role of machine learning-based diagnostic classification in improving clinical decisions. 

Machine learning has shown its potential not only in the diagnostic field but also in dissecting the wide genotypic–phenotypic heterogeneity of ASD and other neurodevelopmental disorders (NDD). Chow and colleagues have used metabolite annotation and gene integration (MAGI)-S, a computational method, to predict modules or groups of highly connected genes that interact to perform similar biological functions [16]. In this case, the aim was to disentangle the epilepsy phenotype from a more general NDD phenotype. Similarly, Peng and colleagues prioritized two modules, enriched in genes associated with both epilepsy and ASD, and coded the biological processes of ion transmembrane transport and synaptic signaling, which may contribute to the shared genetic etiology of epilepsy and ASD. One of the two modules was an epilepsy-focused module enriched in genes directly causing epilepsy and epilepsy phenotypes; the other one was an ASD-focused module enriched in genes related to ASD [3]. 

In a previous study we presented a methodology that made use of hierarchical-agglomerative-clustering, heatmapping, and enrichment analysis [17]. We applied this approach to a freely available database, VariCarta [18], to list and prioritize those biological processes that occur in genetically related clusters of individuals with ASD. The present study builds on more recent statistical and technical developments, with the aim to identify and categorize biological processes that might act as possible pathophysiological substrates for ASD. We propose here a machine learning based approach, which uses genetic data retrieved from VariCarta to evaluate their possible impact on specific ASD endophenotypic characteristics.

## 2. Materials and Methods

### 2.1. Methodological Overview

To identify genetical subtypes of individuals with autism we applied clustering on a pure genetical embedding space, modified to include phenotypical information.

We began by collecting for each individual all the genes related to rare variants. Thereafter, we created a subgroup based on the used sequencing type. We collected only variants retrieved from whole-genome sequencing. We also excluded exome sequencing since it did not perform well neither on clustering in our previous research [17], nor using the present approach based on pre-trained embedding. Then, we projected the gene set of individuals into the genotypical/phenotypical embedding. For each individual we obtained a single vector representation having 64 components. Subsequently, we applied density-based clustering obtaining a set of nine clusters. From each cluster we extracted the set of related genes and applied the enrichment analysis. We also applied some additional analysis to evaluate the impact of the genes on a subset of phenotypes related to ASD. The main elements of the entire process are depicted in Figure 1.

### 2.2. Database

To conduct this research, we used the VariCarta dataset from British Columbia University. It is a web-based database of human DNA genetic variants identified in individuals with an ASD diagnosis. Since all the variants included in VariCarta are collected from ASD genetics research literature, most of them are rare (present in <5% of the population) or very rare (<1% of the population) and only a few are common ones. This information was fundamental for the cluster analysis we carried out. 

VariCarta was developed with the aim to identify rare, possibly causative, genomic variants in individuals with ASD. To tackle this challenge, due to the genetic heterogeneity of ASD, it is necessary to collect a wide variety of individual information through the aggregation of data. This approach can potentially increase the risk of methodological inconsistencies and individual overlaps across studies. VariCarta developers addressed this demanding task by gathering and creating a catalog of literature-derived genomic variants found in individuals with ASD, using an ongoing semi-manual curation and with a robust data import pipeline. Curators, during the continuous development of the database, could identify and correct errors, convert variants into a standardized format, harmonize cohort overlaps, and document data provenance. The VariCarta database is constantly updated with new relevant gene-targeted scientific papers aligned with the ASD research community interests. The current version contains 187,794 variant events from 15,189 individuals, retrieved from 97 papers. The version we used is the one released on 18 May 2022. It consists of 226,495 records, each one containing a variant as reported in the paper from where it was retrieved. Since a single variant belonging to a certain individual and reported in a paper can be reported in other studies as well, we removed duplicated events during the analysis. 

VariCarta dataset is accessible both using a web interface or downloading the whole dataset in csv format. As the web interface allows limited research, we downloaded the whole dataset in csv format. Each row of the dataset corresponds to a variant event which includes, among other information, the symbol of the affected gene, the category of mutation (synonymous SNV and nonsynonymous SNV, frameshift insertion, etc.), the adopted sequencing type (whole genome sequencing, exome sequencing, targeted sequencing), and the individual id that is a unique identifier of the individual presenting the mutation. The dataset also provides references to allow to trace the paper from which the information was collected. Since the number of variants detected in each individual might be affected by the used sequencing type, we handled only whole genome sequencing. In VariCarta the number of variants is revealed by targeted sequencing and exome sequencing is composed, respectively, by 3.0% (5805/187,794 variant events) and 14.1% (26,486/187,794) of all variants. The subset we used related to whole-genome sequencing and forms 84.1% (157,984/187,794 mutations) of all of VariCarta’s reported variants.

### 2.3. Genotypical Embedding Space Creation 

The technique of using embeddings as a vectorial space to identify similarities between elements has been borrowed from the branch of machine learning called natural language processing (NLP). The main insight of this approach is to convert elements (words in this case) into vectors. Assuming that a corpus is composed by a certain number of documents, a word vector can be defined as the number of occurrences of each word in every document so that a word vector would be composed by a number for each document. Since each document represents a dimension of the vectorial space, words having occurrences in the same documents would be closer in the space. This basic approach is called the “bag of words model” [19]. The idea behind the use of these NLP methods in genetics is the replacement of the concept of word with the concept of gene and the creation of a vectorial space that can catch the semantics of “genes language”, i.e., their interactions. In this case, genes interacting with each other should be close in the embedding space.

We used the Gene2Vec [20] as our baseline gene embedding space. Gene2vec developers trained a 200-dimension vector representation of all human genes, using gene co-expression patterns in 984 data sets from the GEO database [21] together with the Gene Ontology [22] resource to identify interactions between genes according to the biological processes they are involved in. These vectors capture functional relatedness of genes in terms of recovering known pathways. Finally, Gensim Python library [23] was used to load the pretrained Gene2Vec embedding and make the subsequent encoding operations.

### 2.4. Phenotypical Embedding Space Creation

To create an embedding space including phenotypical information we combined Gene2Vec with Human Phenotype Ontology (HPO) [24] information (Figure 2). From the HPO we extracted the lists of genes, each impacting on a specific phenotype. For each phenotype, we then created a vector having a dimension for each gene present in the Gene2Vec embedding space (24,447 components) so that every gene always occupies the same dimension. The value of each dimension in a phenotype vector is then zero if the gene is not related to the specific phenotype according to the HPO, otherwise it is equal to the maximum of the 200 components representing the gene in Gene2Vec. The result is a very sparse matrix having as many columns as the number of genes in Gene2Vec and as many rows as the number of phenotypes in HPO. We used an autoencoder having 6 dense layers of encoding and as many dense layers for decoding to reduce the dimensions to 64 components [25]. From VariCarta’s dataset and for each of the individuals included in the subset as defined before, we selected the two features “Gene Symbol” and “Individual id” and generated a sequence of genes for each individual, grouping them by “Individual id”.

We encoded the sequence of genes using the encoder piece of the autoencoder (Figure 3) so that for each individual we obtained a single vector representation having 64 values. The outcoming matrix of the encoded individuals was used for the subsequent clustering step.

### 2.5. Dimensionality Reduction and Clustering 

Dimensionality reduction has been applied to the resulting matrix using uniform manifold approximation and projection (UMAP) for dimension reduction [26]. It allowed the reduction in the dimensions from 64 to 5 to make the computation of the clustering possible. UMAP is an evolution of t-stochastic neighbor embedding (t-SNE) [27] and it is used to obtain a dimensionality reduction that preserves the relative distances between elements (and then their eventual clusters’ structures) going from the original embedding space to the lower dimensional space. The use of UMAP in bioinformatics, particularly in genetics, is not new and it is mainly focused on visualizing multidimensional spaces [28,29].

Finally, the individuals were clustered using hierarchical density-based spatial clustering of applications with noise (HDBSCAN) clustering [30]. On top of our knowledge, HDBSCAN and its not-hierarchical version, called DBSCAN [31], have not been used yet for subtyping individuals with ASD based on genetic variants. To date, the clustering algorithms which are mainly used are agglomerative clustering (bottom-up hierarchical clustering) and K-means [32]. Nevertheless, researchers are beginning to use it in ASD for clustering, based on other features, such as electro-encephalography (EEG) scans [33].

In HDBSCAN, as in other clustering algorithms, the selection hyperparameters play a key role in achieving a high-quality outcome. To select the best hyperparameters, we applied exact grid search cross validation to the following hyperparameters:min_cluster_size: the minimum number of samples a cluster should have. This parameter determines the threshold for a set of samples to be considered as noise.metric: the metric used to measure the distance between samples in the vectorial space. We considered ‘Euclidean’ and ‘Manhattan’.min_samples: the number of neighbors a sample should be close to consider it a cluster sample.cluster_selection_method: the way the clusters are selected in the hierarchy of clusters generated by the algorithm.

To evaluate the clustering results in the cross-validation, we used density-based clustering validation (DBCV) [34]. Another index we considered was the coverage, defined as the ratio between the number of samples belonging to the cluster and the total number of samples. This index provides a clue about the “clusterability” of the data. A low coverage means that most of the samples are marked as noise. A 100% coverage means that no sample has been marked as noise.

### 2.6. Enrichment Analysis and Additional Analyses

Once we identified the set of genes characterizing each cluster of individuals with ASD, we applied to each cluster the enrichment analysis, a methodology used to identify classes of genes or proteins that are over-represented in a large set of genes or proteins and may be associated with specific phenotypes. The analysis was conducted using the Gene Ontology annotation tool (GOAT) [35], a Python library used to simplify the annotation of gene products with terms from the Gene Ontology project. To identify significantly enriched or depleted groups of genes, we compared the input gene set with each of the bins (terms) in the GOAT. The results for each pathway are expressed in terms of fold enrichment (FE), i.e., the ratio between the number of genes in the cluster list belonging to the specific pathway, and the number of genes expected to belong to the pathway in a randomly selected set of genes of the same size. For each gene set we collected the related biological processes, cellular components, and molecular functions. To evaluate the impact of each gene set on phenotypes related to ASD included in the HPO we computed the FE between them and the gene set characterizing each phenotype according to the HPO.

Finally, the gene variants included in the resulting clusters were compared with the human genes associated with ASD, which were retrieved from the Simons Foundation Autism Research Initiative database (SFARI Gene) [36,37]. SFARI Gene is a developing database focusing on genes related to ASD susceptibility (https://gene.sfari.org/), whose data are derived from sources that are in the public domain. Specifically, the Human Gene module of SFARI Gene can be considered an updated reference for known human genes associated with ASD (https://gene.sfari.org/database/human-gene/)(Accessed: 11 January 2023). As of November 2022, the SFARI Gene database contained 1052 genes identified as being ASD-linked.

A conservative statistical significance threshold of *p* < 0.005 (two tailed) was applied for all analyses. We applied the false discovery rate (FDR) using Fisher’s exact test and the Benjamini–Hochberg [38] procedure to control for multiple comparisons. As both raw and FDR-adjusted p-values are strongly dependent on sample size, once the statistically significant terms were identified, we ranked the biological processes by fold enrichment, which, in this context, can be considered a measure of effect size [39].

## 3. Results

### 3.1. Clustering Analysis 

Before applying the clustering, we applied UMAP dimensionality reduction. The following hyperparameters were used:n_neighbors = 15;n_components = 5;Metric = ‘cosine’ distance.

Applying the exact grid search cross validation to HDBSCAN we achieved a coverage of 100% (maximum coverage, i.e., “no noise”) and a DBCV of 0.83. As DBCV ranges from −1 to +1, such a DBCV-value can be considered as high. The metric used in cross-validation is only DBCV so that the full coverage was a good “side-effect” of the optimization. The identified best-fitting hyperparameters were: Min_cluster_size: 105;Metric: ‘Manhattan’ distance;Min_samples: 10;Cluster_selection_method: ‘eom’ (excess of mass).

The total number of individuals belonging to the whole-genome sequencing type group was 3823. The algorithm identified 9 clusters with the largest cluster (cluster 0) including 1455 individuals, while the smallest one (cluster 4) included 106 individuals. The number of variants ranged from 492 (cluster 4) to 17,217 (cluster 0). We then created an intersection between the variants identified in each cluster and the genes that—according to SFARI Gene—are considered ASD-linked. In Table 1, we present the overall results, including the total number of variants and the ASD-linked genes comprised in each cluster. We also enumerated each identified gene variant included in the different clusters and associated it with the corresponding biological pathways and possible ASD phenotype. The extensive register is presented in Appendix A.

To visualize in two dimensions the results arising from the clustering, we further applied UMAP, which reduced the components to two. Figure 4 shows how the nine clusters are distributed into the two-dimensional space.

Additional information related to the density of each cluster in the space is provided by Figure 5. The chart, called a joint plot, looks like an elevation map. 

The condensed tree of the clustering, presented in Figure 6, provides an overview of the behavior of the clustering algorithm. The results of the HDBSCAN are usually strongly influenced by the radius used to bound the density analysis. In the non-hierarchical version of the algorithm, called DBSCAN, the radius must be provided by the user, and it is called the epsilon. It is defined as the maximum distance between two samples, where one sample is considered as being in the neighborhood of the other. In HDBSCAN, the epsilon is not fixed but it is changed by the algorithm to create the cluster’s hierarchy. 

### 3.2. Enrichment Analysis 

We used the set of genes of each cluster for the enrichment analysis. Clusters 4 and 5 did not return any result with an FDR < 0.005. Cluster 0 returned several statistically significant results, but all fold enrichments were <1.5. For completeness, in Table 2 we present the first 20 results from cluster 0, ordered by FDR value. Cluster 1 presented only one result with FE > 1.5 (Table 3). From Table 4, Table 5, Table 6, Table 7 and Table 8 we present the other results ordered by fold enrichment (with FE > 1.5).

Finally, to evaluate the impact of each cluster on the HPO phenotypes related to ASD, we computed the FE between the set of genes belonging to each cluster and the genes related to the phenotype in HPO. The results are presented in Table 9, where we show all the phenotypes with FE > 1.0. Cluster 1 is not present since no FE was above 1.0.

## 4. Discussion

Autism spectrum disorder (ASD) is a clinically heterogeneous neurodevelopmental disorder. The clinical heterogeneity of ASD appears to be closely mirrored by the large variety of ASD-related genes. The genetic architecture of ASD is extremely complex, and it is still an active area of research. Important advancements in the discovery of various molecular mechanisms underlying the genetics of autism and the identification of new ASD risk genes have opened new ways to study the pathophysiology of this disorder [40]. 

Numerous studies have already highlighted the role of different ASD risk genes converging in many biological processes related to various cellular functions, such as gene transcription and translation regulation processes, as well as neuronal activity modulation, synaptic plasticity, disrupted key biological signaling pathways, and ion channels [41,42]. 

Recent advances in ASD understanding have pointed out the role of genotype–phenotype approaches in disentangling the biological bases of the disorder [43]. Indeed, most of the ASD-associated genes can be functionally classified into specific molecular pathways, but it is still a matter of speculation how molecular pathway alterations could affect ASD phenotypes. For example, mouse models have shown how specific abnormal pathways could impact behavioral phenotypes. In mouse models of ASD as well as in clinical neuroscience, behavioral phenotypes, such as impaired social interactions or stereotyped behaviors have been associated with neural circuit dysfunctions and abnormal molecular pathways [44,45]. Similarly, to take another example, our study identified the presence of variants of the CAPRIN1 gene in several clusters, associating it with different biological pathways. CAPRIN 1 was previously related to carcinogenesis [46] and also―in mouse studies― to brain activity and reduced social interaction phenotypes [47]. More recently, loss-of-function variants in this gene have been associated with a neurodevelopmental phenotype presenting, among other characteristics, with language impairment, ADHD, and ASD [48]. It is, therefore, noteworthy to underline that understanding the linkage between ASD genotypes and phenotypes may help to achieve proper diagnosis, predict prognosis, and individualize precision therapy [49].

ASD is likely the result of a complex interaction of factors rather than the consequence of a single factor driving the system. As such, traditional sequencing tools that search for univariate drivers of ASD are unlikely to find consistent patterns. Otherwise, machine learning techniques that explore large search spaces for multivariate interactions are becoming popular in helping to elucidate the complex interactions in systems such as in ASD [13]. Therefore, machine learning approaches have been consistently used as tools for examination, stratification in disease severity, and differential diagnosis in ASD and other neurodevelopmental disorders [13,14,15], as well as for genotype–phenotype studies [3].

Building upon our previous study [17], in this research we used the VariCarta database to identify genetical subgroups of individuals with ASD, applying a novel machine learning approach based on a clustering analysis on a modified embedding space. We obtained different clusters of ASD-related genes and extracted from each cluster the set of related genes. Then, we applied the enrichment analysis to the genes to emphasize crucial biological processes associated with ASD. Finally, we performed an additional analysis to evaluate the impact of these genes on a subset of phenotypes related to ASD.

### 4.1. Cluster Comparisons

Among the nine retrieved gene clusters, two appeared to be of higher clinical relevance (Cluster numbers 2 and 7). Here, biological processes and cellular components related to synaptic communication, such as axon growth and guidance, pre- and post-synaptic membrane components, modulation of chemical synaptic transmission, and post-synaptic density play a fundamental role. These pathways have already been associated with ASD pathogenesis [10,50,51], including in our previous study [17]. Particularly interesting is the fact that some of the processes enriched in Cluster 2 also have a possible direct clinical relevance in terms of phenotypes, as the phenotype fold enrichment per cluster highlighted. This is the case of the CA1 and GABA synapses, which appear to be involved in social interaction difficulties. Indeed, Schaffer collateral-CA1 synapses, potentially linked to hippocampal abnormalities, are crucial for social development and implicated even in autism/epilepsy comorbidity [52]. Regarding GABAergic synapses, the disturbance of the delicate balance between excitation and inhibition in the developing brain profoundly impacts neurobehavioral phenotypes. Analogously, GABA receptor polymorphisms are associated with deficits in social interaction and in sensorimotor and somatosensory coordination, visual response, imitation, and adaptability [53,54]. 

Even the other clusters have shown some possible interesting genotype–phenotype associations. In Cluster 3, for example, the enrichment analysis has put the spotlight on photoreceptor connecting cilium: there is evidence of an altered retinal function in ASD mouse models [55], with consequent atypical visual processing [56]; thus, we can hypothesize a possible association with eye contact deficits due to an impairment of visual sensory processes in ASD, as our phenotype fold enrichment per cluster also suggested.

In Cluster 6, as well as in Cluster 8, ATP binding was pinpointed as a process of interest. According to the literature, mutations in the ATP-binding cassette subfamily A member 13 (ABCA13) have been studied in monkey models for ASD, showing repetitive behaviors [57]. Drosophila models for ASD also showed deficits in social interactions [58]. Both these ASD clinical features have been highlighted by larger fold enrichments in these clusters.

We also compared the number of variants included in each cluster with the number of genes classified by the existing literature as ASD linked. It was not surprising to detect a difference in terms of numbers. In fact, genetic variants identified in an individual or in a group of individuals might be occasional and not necessarily a factor related to the disorder. At the same time, such a difference might also call for the need of studying not only genes but also gene networks and gene interactions as possible ASD causative factors. This is why a key element of this research is the use of a novel machine learning methodology to identify genetic subgroups of individuals with ASD, giving resonance to specific biological processes among different ASD phenotypes. It was used in this study particularly to search for possible links between genetic networks and endophenotypes. 

### 4.2. Translation into Clinical Research

In this study, we implemented patient similarity analysis, which was built upon our earlier work [17] by using a new metric. We used patient similarity algorithms considering that these can play a crucial role in identifying subpopulations of individuals with ASD who could share the same etiopathology. Based on genetic traits or biological activities, molecular processes, and cellular components [59], the identification of subgroups of individuals can be further enhanced. After completing the subgroup categorization process, it is possible to assess the membership of each individual in a particular group by analyzing their distance from the other subgroups. These methods might also help in determining which of the many genetic variations that define ASD [60] play a leading role in contributing to its etiopathology and clinical implications. Additionally, improved approaches could make it possible to distinguish between variations influencing ASD and those influencing other neurodevelopmental disorders.

In the current study, we did not consider targeted analysis sequencing because this technique focuses on the identification of specific genes highly related to a disease, assuming that these are known. However, this assumption can only be made in the case of well-documented “syndromic” ASD, such as, among many others, tuberous sclerosis, Fragile X syndrome, Rubinstein–Taybi syndrome, or Phelan–McDermid syndrome [61,62,63]. Yet, about 85% of all ASD diagnoses are represented by “idiopathic ASD” [64], which might be associated, e.g., with factors such as neuroinflammation, autoimmunity, or metabolic disorders [65,66,67]. Therefore, limiting the analysis to a few genes, while ignoring others could lead to reduced detection of relevant gene variants in a single individual.

### 4.3. Limitations

Several limitations should be kept in mind when interpreting our findings. First off, we did not conduct the same research on a sample of people without ASD, as VariCarta does not include data about these individuals. This restriction does not allow to distinguish between de novo mutations and those found in the genomes of the biological parents [50]. Likewise, population stratification analysis was not feasible since VariCarta does not disclose any information about age, gender, ethnicity, family relationships, or other personal characteristics of the included individuals. Furthermore, VariCarta does not include details on each person’s homozygous/heterozygous status. Hence, the variability of the impacts of the variations connected to this characteristic could not be evaluated in the current analysis, even though, given the design of this study, the absence of this information would have probably had a negligible effect on the results. A further limitation, related to the characteristics of the used dataset, concerns the inability to remove common variants and evaluate variant deleteriousness in the cluster analysis. However, from analyses conducted in our previous work [17], this appears to be a minor limitation. It must be emphasized that we used a database that excluded environmental or epigenetic variables, restricting the classification of the subgroups exclusively to genetic variant events. Additionally, all variations were included in the study; there was no selection based on variation nature (base substitution, deletion, or insertion), category of nucleotide variation, or category of sequence variation (exonic or intronic). Even though the variations included in VariCarta were obtained from published controlled studies where the variant was related with ASD, it is possible that not at all genetic variants were implicated in ASD. Finally, even though the proposed methodology allowed us to assess the impact of variants on a subset of phenotypes related to ASD and epilepsy, and preliminary assessments conducted on the clusters were found in the literature for some possible genotype–phenotype associations, the absence in the dataset of the description of each phenotype, including gender and IQ, did not allow us to confirm our clustering results.

In conclusion, since ASD is a multigenic and highly heterogeneous condition, innovative methodologies, including machine learning and newly developed biomedical informatics, can improve our understanding of the underlying biological processes that undergo the etiology and the pathogenic mechanisms of ASD and may identify more homogeneous subgroups of individuals with ASD. Due to the complex architecture of ASD, similarity analysis and machine learning might be helpful in forecasting developmental trajectories [68], offering therapeutic decision assistance [69], and customizing individual therapies [70]. The methodology experimented here in the context of ASD could also be a promising tool for the study of other disorders. Nonetheless, further research comparing the identified biological processes, the shared genetic pathways, and the convergent endophenotypes with associated phenotypes will be necessary to confirm the clinical validity and usefulness of our results. 

## Figures and Tables

**Figure 1 genes-14-00313-f001:**
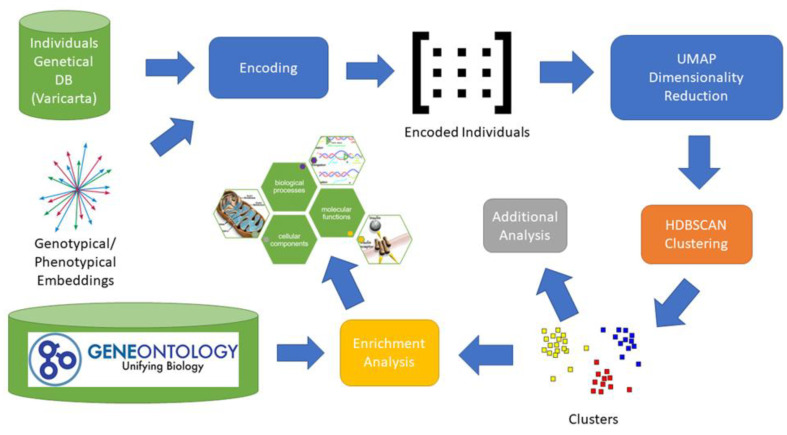
Analysis Process. The image depicts the entire process adopted to identify potential subgroups of individuals with ASD. For each individual included in the VariCarta database, the set of variated genes is selected and then encoded using the genotypical/phenotypical embedding space. Each individual is then represented with a 64-component vector in the genotypical/phenotypical embedding space. Dimensionality reduction is applied to the encoded individuals’ matrix to reduce clustering complexity. The genes of the resulting clusters are then used for the enrichment and endophenotype analysis.

**Figure 2 genes-14-00313-f002:**
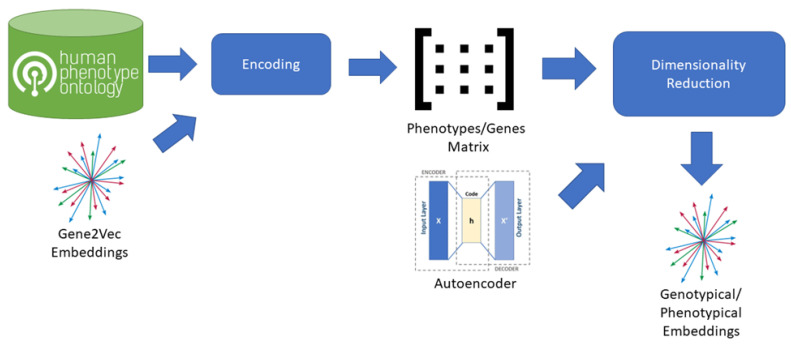
Process adopted to generate Genotypical/Phenotypical Embeddings from Gene2Vec. The process starts from a pre-existing embedding space for human genome that is Gene2Vec. Gene2Vec captures all the semantics of the interactions between genes, meaning that two genes are close in the embedding space if they have a mutual string interaction. For each phenotype in the HPO database the set of characterizing genes is extracted and encoded using Gene2Vec. The encoding transforms each gene into a 200-component vector. From the encoded phenotypes, a phenotypes/genes matrix is composed, having as many columns as the number of genes and as many rows as the number of phenotypes. Dimensionality reduction is applied using an autoencoder to reduce the initial 24,447 components to 64 components.

**Figure 3 genes-14-00313-f003:**
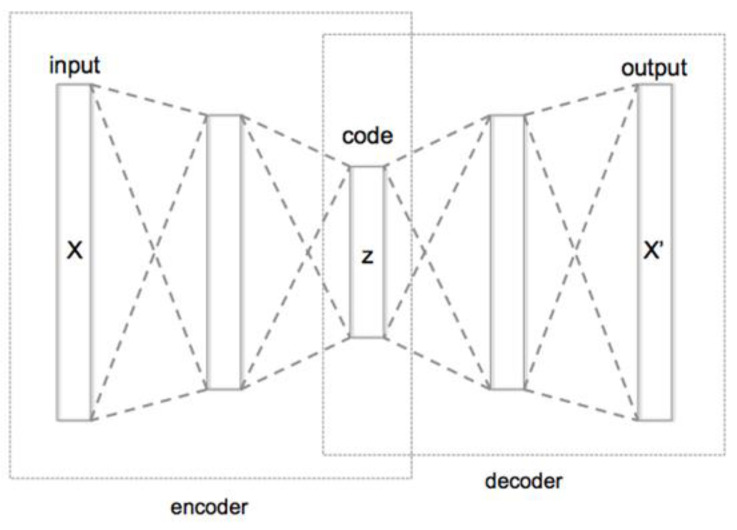
Base structure of a Deep Autoencoder for dimensionality reduction. The autoencoder is a deep learning structure usually composed of an encoder component and a subsequent decoder component. To generate a representation of some data X in the form of an embedding, the autoencoder is trained to reproduce X. This means that the loss of the training is computed between the decoder output X’ and the input X. The purpose is to reproduce an output that is as similar as possible to the input. Once this goal is reached with the desired level of accuracy, it means that the decoder can properly reproduce the data from the encoder representation z, which is usually a lower dimension version of the input data X.

**Figure 4 genes-14-00313-f004:**
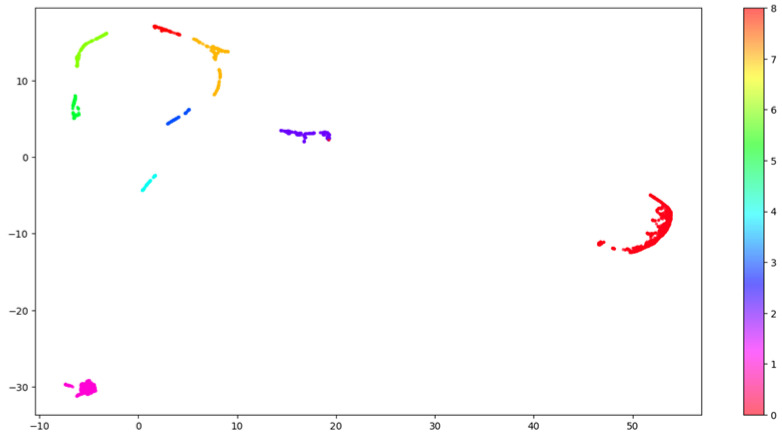
HDBSCAN clustering. This image shows the distribution of the clusters in the embedding space. The original embedding space including 64 dimensions was compressed into 2 dimensions using the UMAP algorithm to allow 2-dimensional visualization. Each one of the nine clusters is labelled using a distinct color.

**Figure 5 genes-14-00313-f005:**
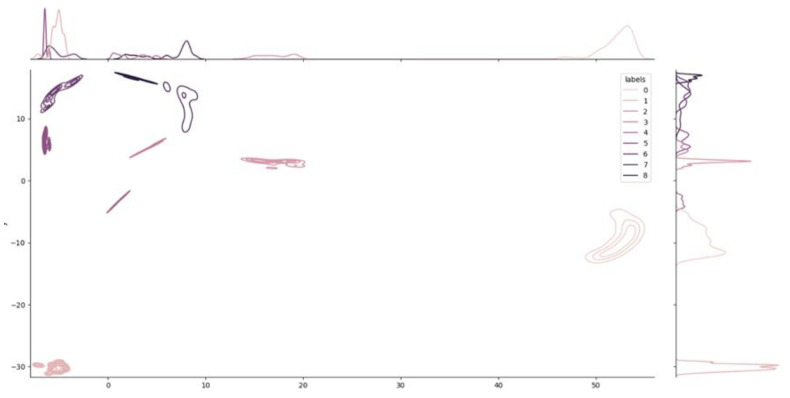
Joint Plot and density distribution of the clusters. Like an elevation map, the joint plot shows the local density of each cluster. The two plots on the two axes show the decomposition of the density into the two dimensions used for visualization. The dimensionality reduction from 64 to 2 dimensions was obtained using the UMAP algorithm.

**Figure 6 genes-14-00313-f006:**
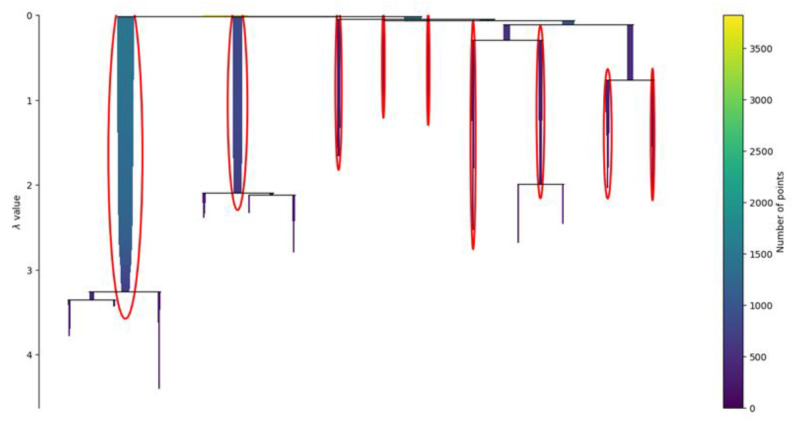
Condensed tree. The condensed tree provides a view of the behavior of the clustering algorithm. In the ordinate, the parameter lambda represents the inverse of epsilon, defined as the maximum distance between two samples, where one sample is considered as being in the neighborhood of the other. The root of the hierarchy is where the value of lambda is small, which means that the epsilon distance is wide. In this area, the identified clusters are larger since the definition of neighbor is wider. Once lambda increases and epsilon decreases, the clusters are sliced into smaller clusters. Usually, a robust clustering is considered the one that persists despite the large variations of lambda. The clusters circled in red are the nine ones selected by the algorithm and are the ones with higher persistence.

**Table 1 genes-14-00313-t001:** Number of individuals, genetic variants, and ASD-linked genes included in each cluster.

CLUSTER INDEX	INDIVIDUALS	VARIANTS	ASD-LINKED GENES *
0	1455	17,217	879
1	841	1747	154
2	273	7509	516
3	110	558	49
4	106	492	41
5	214	944	96
6	334	1859	188
7	336	5296	410
8	154	1186	117

* Based on data contained within SFARI Gene as of November 2022. The nine clusters determined by the algorithm are presented according to three characteristics: number of individuals, number of variants, and number of variants, which, according to SFARI Gene, are considered ASD-linked. An additional table in the Appendix A shows the full list of gene variants and of ASD-linked genes included in each cluster, as well as biological pathways and phenotypes related to these variants (Appendix A).

**Table 2 genes-14-00313-t002:** Enrichment Analysis for Cluster 0.

GO Element Type	GO Code	GO Name	FE	FDR
molecular_function	GO:0005515	Protein binding	1.087222547	1.55 × 10^−99^
cellular_component	GO:0005886	Plasma membrane	1.147275591	2.24 × 10^−55^
cellular_component	GO:0005737	Cytoplasm	1.130739139	4.92 × 10^−47^
cellular_component	GO:0005829	Cytosol	1.121747304	5.30 × 10^−46^
molecular_function	GO:0005524	ATP binding	1.224015929	2.82 × 10^−35^
molecular_function	GO:0046872	Metal ion binding	1.161264333	4.55 × 10^−29^
cellular_component	GO:0005654	Nucleoplasm	1.116129667	2.34 × 10^−27^
cellular_component	GO:0000786	Nucleosome	0.299454744	3.29 × 10^−25^
cellular_component	GO:0005634	Nucleus	1.083234714	1.05 × 10^−21^
cellular_component	GO:0005794	Golgi apparatus	1.20614718	1.76 × 10^−20^
cellular_component	GO:0016020	Membrane	1.128252261	6.39 × 10^−17^
molecular_function	GO:0004712	Protein serine/threonine/tyrosine kinase activity	1.290737468	1.18 × 10^−16^
cellular_component	GO:0043231	Intracellular membrane-bounded organelle	1.198508723	3.19 × 10^−16^
biological_process	GO:0006334	Nucleosome assembly	0.373973889	5.38 × 10^−16^
cellular_component	GO:0005887	Integral component of plasma membrane	1.148641895	1.45 × 10^−14^
molecular_function	GO:0004674	Protein serine/threonine kinase activity	1.29806618	1.69 × 10^−14^
molecular_function	GO:0106310	Protein serine kinase activity	1.294450396	2.73 × 10^−14^
cellular_component	GO:0098978	Glutamatergic synapse	1.297368237	2.94 × 10^−13^
biological_process	GO:0006468	Protein phosphorylation	1.250748447	3.98 × 10^−13^
cellular_component	GO:0030424	Axon	1.289923348	7.28 × 10^−13^

GO: Gene Ontology; FE: fold enrichment; FDR: false discovery rate *p*-value. Biological processes, molecular functions, and cellular components are identified by their reference numbers (GO:XXXXXX) in Gene Ontology. Results with FDR < 0.005 and FE < 1.5 are shown. Results, ranked by FDR, are shown up to the 20th value. An additional table in the Appendix A shows a full list of the 286 biological processes, molecular functions, and cellular components (Appendix A).

**Table 3 genes-14-00313-t003:** Enrichment Analysis for Cluster 1.

GO Element Type	GO Code	GO Name	FE	FDR
cellular_component	GO:0005886	Plasma membrane	1.246675801	1.52 × 10^−4^

FE: fold enrichment; FDR: false discovery rate p-value. Biological processes, molecular functions, and cellular _components are identified by their reference numbers (GO:XXXXXX) in Gene Ontology. A single result with FDR < 0.005 and FE <1.5 was obtained. An additional table in the Appendix A shows a list of the 3 molecular functions and cellular components (Appendix A).

**Table 4 genes-14-00313-t004:** Enrichment Analysis for Cluster 2.

GO Element Type	GO Code	GO Name	FE	FDR
biological_process	GO:0006939	Smooth muscle contraction	2.82253091	4.97 × 10^−3^
molecular_function	GO:0005001	Transmembrane receptor protein tyrosine phosphatase activity	2.82253091	2.50 × 10^−3^
cellular_component	GO:0016342	Catenin complex	2.763999163	8.48 × 10^−7^
molecular_function	GO:1904315	Transmitter-gated ion channel activity involved in regulation of postsynaptic membrane potential	2.520116883	5.00 × 10^−5^
biological_process	GO:0060078	Regulation of postsynaptic membrane potential	2.513396572	6.85 × 10^−4^
cellular_component	GO:0044295	Axonal growth cone	2.492624699	1.75 × 10^−3^
biological_process	GO:0098742	Cell–cell adhesion via plasma-membrane adhesion molecules	2.408414405	3.11 × 10^−4^
molecular_function	GO:0043325	Phosphatidylinositol-3,4-bisphosphate binding	2.363695836	2.79 × 10^−3^
cellular_component	GO:0099061	Integral component of postsynaptic density membrane	2.150499741	1.16 × 10^−4^
cellular_component	GO:0098839	Postsynaptic density membrane	2.089853025	2.01 × 10^−3^
biological_process	GO:0050804	Modulation of chemical synaptic transmission	2.068233856	1.14 × 10^−3^
biological_process	GO:0051056	Regulation of small GTPase-mediated signal transduction	1.996809134	5.33 × 10^−7^
molecular_function	GO:0008013	β-catenin binding	1.992650559	1.82 × 10^−5^
cellular_component	GO:0031594	Neuromuscular junction	1.965691169	1.29 × 10^−4^
cellular_component	GO:0098982	GABA-ergic synapse	1.943875232	1.98 × 10^−4^
cellular_component	GO:0042734	Presynaptic membrane	1.924131347	1.54 × 10^−3^
biological_process	GO:0043087	Regulation of GTPase activity	1.904088312	7.68 × 10^−4^
biological_process	GO:0007411	Axon guidance	1.792469342	6.04 × 10^−7^
biological_process	GO:0006470	Protein dephosphorylation	1.771967898	3.42 × 10^−5^
cellular_component	GO:0045211	Postsynaptic membrane	1.764081818	9.26 × 10^−6^
cellular_component	GO:0098685	Schaffer collateral-CA1 synapse	1.761281689	3.45 × 10^−3^
cellular_component	GO:0098978	Glutamatergic synapse	1.713679481	3.42 × 10^−12^
cellular_component	GO:0042383	Sarcolemma	1.660679084	3.34 × 10^−3^
molecular_function	GO:0017124	SH3 domain binding	1.658399498	2.01 × 10^−3^
biological_process	GO:0009887	Animal organ morphogenesis	1.65020987	2.70 × 10^−3^
biological_process	GO:0007420	Brain development	1.628703639	4.48 × 10^−6^
cellular_component	GO:0005938	Cell cortex	1.626910899	1.29 × 10^−4^
biological_process	GO:0098609	Cell–cell adhesion	1.608369569	5.42 × 10^−4^
cellular_component	GO:0005912	Adherens junction	1.603442789	1.16 × 10^−4^
molecular_function	GO:0005085	Guanyl nucleotide exchange factor activity	1.591273804	2.63 × 10^−5^
cellular_component	GO:0014069	Postsynaptic density	1.58515352	4.25 × 10^−6^
biological_process	GO:0007268	Chemical synaptic transmission	1.574355946	4.57 × 10^−5^
cellular_component	GO:0030054	Cell junction	1.556385229	7.80 × 10^−5^
cellular_component	GO:0030424	Axon	1.555005455	1.13 × 10^−7^
cellular_component	GO:0043005	Neuron projection	1.550471911	1.13 × 10^−7^
biological_process	GO:0016477	Cell migration	1.545671689	1.07 × 10^−4^
cellular_component	GO:0043197	Dendritic spine	1.54344642	2.41 × 10^−3^
cellular_component	GO:0042995	Cell projection	1.542311533	3.45 × 10^−3^
cellular_component	GO:0045202	Synapse	1.525092744	7.81 × 10^−9^
cellular_component	GO:0030425	Dendrite	1.511341422	2.49 × 10^−8^
molecular_function	GO:0005516	Calmodulin binding	1.508037943	2.77 × 10^−3^
biological_process	GO:0007399	Nervous system development	1.502114113	4.57 × 10^−5^
molecular_function	GO:0031267	Small GTPase binding	1.501876399	1.15 × 10^−4^

GO: Gene Ontology; FE: fold enrichment; FDR: false discovery rate *p*-value. Biological processes, molecular functions, and cellular components are identified by their reference numbers (GO:XXXXXX) in Gene Ontology. Results with FE ≥ 1.5 and FDR < 0.005 are selected and ranked by FE. An additional table in the Appendix A shows a full list of the 149 biological processes, molecular functions, and cellular components (Appendix A).

**Table 5 genes-14-00313-t005:** Enrichment Analysis for Cluster 3.

GO Element Type	GO Code	GO Name	FE	FDR
cellular_component	GO:0032391	Photoreceptor connecting cilium	8.843272901	1.02 × 10^−3^

GO: Gene Ontology; FE: fold enrichment; FDR: false discovery rate *p*-value. Biological processes, molecular functions, and cellular components are identified by their reference numbers (GO:XXXXXX) in Gene Ontology. A single result with FE ≥ 1.5 and FDR < 0.005 is shown. An additional table in the Appendix A shows a list of the cellular components (Appendix A).

**Table 6 genes-14-00313-t006:** Enrichment Analysis for Cluster 6.

GO Element Type	GO Code	GO Name	FE	FDR
molecular_function	GO:0005516	Calmodulin binding	2.394767442	9.63 × 10^−4^
cellular_component	GO:0030424	Axon	1.983969128	2.02 × 10^−3^
molecular_function	GO:0005524	ATP binding	1.548471524	1.05 × 10^−5^
cellular_component	GO:0005886	Plasma membrane	1.271435899	5.21 × 10^−6^

GO: Gene Ontology; FE: fold enrichment; FDR: false discovery rate *p*-value. Biological processes, molecular functions, and cellular components are identified by their reference numbers (GO:XXXXXX) in Gene Ontology. Three results with FE ≥ 1.5 and FDR < 0.005 are shown. An additional table in the Appendix A shows a full list of the 12 biological processes, molecular functions, and cellular components (Appendix A).

**Table 7 genes-14-00313-t007:** Enrichment Analysis for Cluster 7.

GO Element Type	GO Code	GO Name	FE	FDR
molecular_function	GO:0008066	Glutamate receptor activity	4.18111949	2.10 × 10^−5^
biological_process	GO:0007413	Axonal fasciculation	3.520942728	2.12 × 10^−3^
molecular_function	GO:0098632	Cell–cell adhesion mediator activity	3.185614849	2.38 × 10^−4^
molecular_function	GO:0050840	Extracellular matrix binding	3.026334107	3.24 × 10^−4^
cellular_component	GO:0016342	Catenin complex	2.774567772	1.13 × 10^−3^
biological_process	GO:0050804	Modulation of chemical synaptic transmission	2.553984319	3.06 × 10^−4^
cellular_component	GO:0099061	Integral component of postsynaptic density membrane	2.248669305	3.79 × 10^−3^
cellular_component	GO:0005912	Adherents junction	2.123743233	2.53 × 10^−9^
biological_process	GO:0051056	Regulation of small GTPase-mediated signal transduction	1.994471906	1.25 × 10^−3^
biological_process	GO:0018108	Peptidyl-tyrosine phosphorylation	1.978565473	6.13 × 10^−4^
biological_process	GO:0007411	Axon guidance	1.894891591	2.12 × 10^−4^
cellular_component	GO:0030424	Axon	1.858275329	9.42 × 10^−11^
molecular_function	GO:0005201	Extracellular matrix structural constituent	1.848586719	9.09 × 10^−4^
molecular_function	GO:0008017	Microtubule binding	1.810574065	1.12 × 10^−6^
cellular_component	GO:0045211	Postsynaptic membrane	1.791908353	1.55 × 10^−3^
biological_process	GO:0007156	Homophilic cell adhesion via plasma membrane adhesion molecules	1.784711934	2.12 × 10^−3^
molecular_function	GO:0051015	Actin filament binding	1.73356715	1.40 × 10^−4^
molecular_function	GO:0005516	Calmodulin binding	1.720232019	3.71 × 10^−4^
molecular_function	GO:0003779	Actin binding	1.719508015	1.91 × 10^−5^
cellular_component	GO:0098978	Glutamatergic synapse	1.643533776	8.85 × 10^−6^
cellular_component	GO:0043235	Receptor complex	1.637052075	8.44 × 10^−4^
biological_process	GO:0007420	Brain development	1.61913482	4.92 × 10^−3^
molecular_function	GO:0005096	GTPase activator activity	1.598663334	5.70 × 10^−4^

GO: Gene Ontology; FE: fold enrichment; FDR: false discovery rate *p*-value. Biological processes, molecular functions, and cellular components are identified by their reference numbers (GO:XXXXXX) in Gene Ontology. Results with FE ≥ 1.5 and FDR < 0.005 are selected and ranked by FE. An additional table in the Appendix A shows a full list of the 90 biological processes, molecular functions, and cellular components (Appendix A).

**Table 8 genes-14-00313-t008:** Enrichment Analysis for Cluster 8.

GO Element Type	GO Code	GO Name	FE	FDR
molecular_function	GO:0004712	Protein serine/threonine/tyrosine kinase activity	2.162900762	1.88 × 10^−3^
molecular_function	GO:0005524	ATP binding	1.716714944	1.24 × 10^−5^

GO: Gene Ontology; FE: fold enrichment; FDR: false discovery rate *p*-value. Biological processes, molecular functions, and cellular components are identified by their reference numbers (GO:XXXXXX) in Gene Ontology. Two results with FE ≥ 1.5 and FDR < 0.005 are selected and ranked by FE. An additional table in the Appendix A shows a full list of the 7 biological processes and molecular functions (Appendix A).

**Table 9 genes-14-00313-t009:** Phenotype Fold Enrichment by Cluster.

Cluster/ Phenotype	0	2	3	4	5	6	7	8
Restrictive behavior		1.22		2.33		1.23		1.93
Impaired social interactions	1.05	1.16	1.12	3.81	1.32	2.01	1.18	
Poor eye contact			1.55		1.15	1.40		2.56
Lack of peer relationships	1.20	1.65		4.16		2.22	1.17	1.74
Restrictive behavior						1.23		
Impaired ability to form peer relationships	1.20	1.83		13.95				
Abnormal non-verbal communicative behavior		1.10		8.37				

FE: fold enrichment. Genes belonging to each cluster and the genes related to the phenotype in HPO. Results are shown ranked by fold enrichment. An additional table in the Appendix A shows the full list of phenotypes of interest (Appendix A).

## Data Availability

The VariCarta dataset is freely available at https://varicarta.msl.ubc.ca/index, both using a web interface or by downloading the whole dataset in csv format. SFARI Gene data are derived from sources that are in the public domain and are freely available at https://gene.sfari.org/ (accessed on 11 January 2023). All data generated or analyzed during the present study are included in this published article and its Appendix A.

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
