# Peer review of "Using Machine Learning to Explore Shared Genetic Pathways and Possible Endophenotypes in Autism Spectrum Disorder"

_genes, 2023, doi:10.3390/genes14020313_

Round 1
Reviewer 1 Report
The article is very well written and it is quite innovative. Indeed, the study uses bioinformatics tools depicting genetic pathways related generally to autism spectrum disorders as well as those more connected to specific autistic features in a multivariate approach, resembling the multifactorial origin of these disorders. The authors are also honest in describing the limitations of their work. As authors themselves write, studies similar to this one could help predict prognosis, and customize therapy in ASD, although not always in a specific pattern. This article enriches the literature of these new insights, especially to encourage the advances of other studies in this field.
However, one major revision should be performed:
- The study has a great potential, yet it does not fully describe all genes included in the shown pathways, which are recurrent in ASD. Therefore, it would be interesting to add a supplementary table listing the most recurrent genes related to autistic features, beside the examples reported in the discussion.
Minor revisions:
- I suggest avoiding some pleonastic concepts, such as:
- Introduction, page 1, line 42-44 and page 2 line 59-60;
- Materials and Methods: Methodological overview, page 3 line 102, and Database, page 4 line 158;
It could be more suitable to recall some concepts in Discussion.
- Discussion, page 15, line 508-510 and line 524-526. - Page 10, line 361, typo: please change “Custer” into “Cluster”.
Author Response
Point by point description of the revisions to the manuscript and responses to the referees’ comments
We have accepted all the suggestions of the reviewers and revised the manuscript accordingly. All revisions are highlighted in the manuscript in red font.
Reviewer 1
The article is very well written and it is quite innovative. Indeed, the study uses bioinformatics tools depicting genetic pathways related generally to autism spectrum disorders as well as those more connected to specific autistic features in a multivariate approach, resembling the multifactorial origin of these disorders. The authors are also honest in describing the limitations of their work. As authors themselves write, studies similar to this one could help predict prognosis, and customize therapy in ASD, although not always in a specific pattern. This article enriches the literature of these new insights, especially to encourage the advances of other studies in this field.
We thank the reviewer for this comment. We are aware of possible limitations, but the reviewer’s words encourage us to keep on working in this direction.
However, one major revision should be performed: The study has a great potential, yet it does not fully describe all genes included in the shown pathways, which are recurrent in ASD. Therefore, it would be interesting to add a supplementary table listing the most recurrent genes related to autistic features, beside the examples reported in the discussion.
We deeply appreciate this suggestion. In fact, the indication of linking the results of our study to the known ASD-linked genes is an excellent one. So, following the reviewer’s suggestion, we now connected the genes of our clusters with the ones present in the SFARI Genes database. We believe that the obtained results are adding value to our research (please, see lines 273-282, 310-313, Table 1, and 463-466). We added also a complete Table in the Supplementary Materials (Supplementary Table 1), where all the ASD-linked genes included in our clusters are presented. Thank you very much indeed.
Minor revisions:
I suggest avoiding some pleonastic concepts, such as:
- Introduction, page 1, line 42-44 and page 2 line 59-60;
- Materials and Methods: Methodological overview, page 3 line 102, and Database, page 4 line 158.
We removed several redundant concepts throughout the manuscript, in the Introduction Methodology, and Results. We just kept in the very beginning of the Introductions the first sentence about ASD. As Genes is a journal read by a wide and variegated audience, we believe that a sentence to introduce ASD might be appreciated.
It could be more suitable to recall some concepts in Discussion.
- Discussion, page 15, line 508-510 and line 524-526.
The reviewer is right. Some concepts have been now moved to the Discussion or conclusions (please see lines 462-465, 540-555, 595-600).
Page 10, line 361, typo: please change “Custer” into “Cluster”.
We corrected the typo.
Reviewer 2 Report
General comments: Thank you for giving me the opportunity to review this comprehensive and structured manuscript. The authors have clearly provided detailed methods of the study procedures with adequate data interpretation. Additionally, they have explicitly demonstrated the limitations of their study with clear directions for future research. However, I have some comments that would help in improving this manuscript.
Introduction:
It is well written, providing clear, sufficient background information. I just had few comments.
In line 45: Consider revising this sentence
In line 59: “ASD can cause significant social, communication……” I suggest editing this sentence as ASD is characterized by these deficits not it causes them.
In line 89: It would be helpful to provide briefly about the methods and findings of this previous study.
In line 91: I suggest adding the reference for the VariCarta database here as well.
Results: This section is clearly written and described in detail.
Materials and Methods
In page 7: I am not sure if the table is complete or not as there is heading in the last column without data underneath. Also, I would suggest changing the title for table 1 to better reflect its contents.
In line 330: I suggest using provided instead of by.
In line 432: I suggest changing “cluster-by-cluster of the interested phenotypes” to phenotypes of interest.
Discussion
In line 454: In this sentence “could impact on behavioral phenotypes”, I suggest removing “on”. Also, it would be helpful to elaborate more on these behavioral phenotypes.
In line 489” In this sentence “profoundly impacts on neurobehavioral phenotypes”, I suggest removing “on”.
It will be helpful to clarify more on implications of this study on future research and how this work be expanded in the future.
Author Response
Point by point description of the revisions to the manuscript and responses to the referees’ comments
We have accepted all the suggestions of the reviewers and revised the manuscript accordingly. All revisions are highlighted in the manuscript in red font.
Reviewer 2
General comments: Thank you for giving me the opportunity to review this comprehensive and structured manuscript. The authors have clearly provided detailed methods of the study procedures with adequate data interpretation. Additionally, they have explicitly demonstrated the limitations of their study with clear directions for future research. However, I have some comments that would help in improving this manuscript.
Thank for your valuable comments, which helped us to improve the quality of the manuscript.
Introduction: It is well written, providing clear, sufficient background information. I just had few comments.
In line 45: Consider revising this sentence.
We thank the reviewer for his suggestion, we have rephrased the sentence.
In line 59: “ASD can cause significant social, communication……” I suggest editing this sentence as ASD is characterized by these deficits not it causes them.
We have rewritten the sentence based on the reviewer's suggestions.
In line 89: It would be helpful to provide briefly about the methods and findings of this previous study.
We have modified the sentence adding information about the methodology we have presented in our previous work, and about the findings we obtained (please, see lines 84-88).
In line 91: I suggest adding the reference for the VariCarta database here as well.
As suggested, we have added the reference.
Results: This section is clearly written and described in detail.
Thank you.
Materials and Methods
In page 7: I am not sure if the table is complete or not as there is heading in the last column without data underneath. Also, I would suggest changing the title for table 1 to better reflect its contents.
We fixed the formatting of Table 1 and changed the title trying to make it more immediate and readable.
In line 330: I suggest using provided instead of by.
The sentence was changed accordingly.
In line 432: I suggest changing “cluster-by-cluster of the interested phenotypes” to phenotypes of interest.
We have rephrased the sentence following the reviewer suggestion.
In line 454: In this sentence “could impact on behavioral phenotypes”, I suggest removing “on”. Also, it would be helpful to elaborate more on these behavioral phenotypes.
We have deleted “on” from the sentence and tried to better discuss the concept of behavioural phenotypes, by adding also new references (please, see lines 464-467).
In line 489” In this sentence “profoundly impacts on neurobehavioral phenotypes”, I suggest removing “on”.
As suggested, we have removed “on” from the sentence.
It will be helpful to clarify more on implications of this study on future research and how this work be expanded in the future.
We expanded the conclusions by underscoring possible implications and future expansions of the study (please, see lines 594-600).
Reviewer 3 Report
The work presented is excellent, clear and of great contribution to the scientific community that studies the pathophysiology of ASD. Being this very heterogeneous disorder with the urgent need to obtain "biological markers" for a correct molecular diagnosis that allows us to subdivide into groups or endophenotypes.
It is important to highlight two aspects:
1) "Secondary Autism" is not only related to X-fragile syndrome or tuberous sclerosis. There are other syndromes such as RTS, Phelan-McDermid, channelopathies, among others.
2) The so-called "ideopathic autism" that many researchers want to leave in that term is rather related to other causes published in the literature such as: neuroinflammation, autoimmunity, metabolic disorders, inborn errors of metabolism, deregulation of the gut-brain axis and some recently published aspects such as the differential diagnosis of autism and post-traumatic stress disorder.
These aspects mentioned above, added to the extremely complicated genetic architecture of autism, make this type of study an exceptionally valuable and necessary path in these times. My congratulations.
P.S. I am available to contribute with cases from Latin America in the future.
Author Response
Point by point description of the revisions to the manuscript and responses to the referees’ comments
We have accepted all the suggestions of the reviewers and revised the manuscript accordingly. All revisions are highlighted in the manuscript in red font.
Reviewer 3
The work presented is excellent, clear and of great contribution to the scientific community that studies the pathophysiology of ASD. Being this very heterogeneous disorder with the urgent need to obtain "biological markers" for a correct molecular diagnosis that allows us to subdivide into groups or endophenotypes.
Thank you. We appreciate your encouragement to continue researching in this direction.
It is important to highlight two aspects:
1) "Secondary Autism" is not only related to X-fragile syndrome or tuberous sclerosis. There are other syndromes such as RTS, Phelan-McDermid, channelopathies, among others.
The reviewer is right. We better clarified this concept in the Discussion and added some references (please, see lines 540-548).
2) The so-called "ideopathic autism" that many researchers want to leave in that term is rather related to other causes published in the literature such as: neuroinflammation, autoimmunity, metabolic disorders, inborn errors of metabolism, deregulation of the gut-brain axis and some recently published aspects such as the differential diagnosis of autism and post-traumatic stress disorder.
Thank you. We now expanded the concept of idiopathic autism and added some references in the Discussion (please, see lines 550-555).
These aspects mentioned above, added to the extremely complicated genetic architecture of autism, make this type of study an exceptionally valuable and necessary path in these times. My congratulations.
P.S. I am available to contribute with cases from Latin America in the future.
We would be more than happy to cooperate in future, by including cases from Latin America!
Round 2
Reviewer 1 Report
Authors have satisfied the suggested revisions, also providing the ASD-linked genes included in their clusters. I suggest only to link ASD-linked genes of Supplementary Table 1 with their corresponding biological pathway (reported in Supplementary Table 2) and, if possible, with ASD phenotypes (in Supplementary Table 3). It would be really interesting and useful for researchers.
Author Response
Point by point description of the revisions to the manuscript and response to the comment of Reviewer 1
We have accepted the suggestion of the Reviewer 1 and revised the manuscript accordingly. The revision consists in adding new pieces of information to Supplementary Table 1 (four more sheets) and a brief description in the manuscript of the expanded Supplementary Table 1. The revision in the manuscript is highlighted in red font.
Reviewer 1
Authors have satisfied the suggested revisions, also providing the ASD-linked genes included in their clusters. I suggest only to link ASD-linked genes of Supplementary Table 1 with their corresponding biological pathway (reported in Supplementary Table 2) and, if possible, with ASD phenotypes (in Supplementary Table 3). It would be really interesting and useful for researchers.
We thank the Reviewer for his/her positive comment after our first revision.
We deeply appreciate the Reviewer’s new proposal. The hint of linking ASD-linked genes of Supplementary Table 1 to their corresponding biological pathways and, if possible, with ASD phenotypes was actually stimulating.
Extensive and time-consuming work was required to reconstruct the correspondence between genes and pathways and between genes and phenotypes. But we believe that the effort was fully justified by the updated results, which we consider of great interest both because they enrich the analysis and because they can potentially open the way for new research on genes of potential interest to ASD. Thank you very much indeed.
Thus, following the Reviewer’s suggestion, we added in the Supplementary Materials an updated version of Supplementary Table A1, where we now connected the genes of our clusters with their corresponding biological pathway (Supplementary Table A1, sheet “Genes and related Pathways”) and then the ones present in the SFARI Genes database (Supplementary Table 1, sheet “ASD-linked genes and Pathways”).
Furthermore, always following the suggestion received, we connected the genes of our clusters with ASD phenotypes (Supplementary Table 1, sheet “Genes and related Phenotypes”) and then the ones present in the SFARI Genes database (Supplementary Table 1, sheet “ASD-linked genes and Phenotypes”).
In relation to the manuscript, please see lines 312-314, caption of Table 1, and 605-607.